# Microwave-Based Technique for Fast and Reliable Extraction of Organic Contaminants from Food, with a Special Focus on Hydrocarbon Contaminants

**DOI:** 10.3390/foods8100503

**Published:** 2019-10-16

**Authors:** Sabrina Moret, Chiara Conchione, Ana Srbinovska, Paolo Lucci

**Affiliations:** Department of Agri-Food, Environmental and Animal Sciences, University of Udine, 33100 Udine, Italy; chiara.conchione@uniud.it (C.C.); srbinovska.ana@spes.uniud.it (A.S.); paolo.lucci@uniud.it (P.L.)

**Keywords:** food contaminants, polycyclic aromatic hydrocarbons (PAH), mineral oil hydrocarbons (MOH), microwave-assisted extraction (MAE), microwave-assisted saponification (MAS)

## Abstract

Due to food complexity and the low amount at which contaminants are usually present in food, their analytical determination can be particularly challenging. Conventional sample preparation methods making use of large solvent volumes and involving intensive sample manipulation can lead to sample contamination or losses of analytes. To overcome the disadvantages of conventional sample preparation, many researchers put their efforts toward the development of rapid and environmental-friendly methods, minimizing solvent consumption. In this context, microwave-assisted-extraction (MAE) has obtained, over the last years, increasing attention from analytical chemists and it has been successfully utilized for the extraction of various contaminants from different foods. In the first part of this review, an updated overview of the microwave-based extraction technique used for rapid and efficient extraction of organic contaminants from food is given. The principle of the technique, a description of available instrumentation, optimization of parameters affecting the extraction yield, as well as integrated techniques for further purification/enrichment prior to the analytical determination, are illustrated. In the second part of the review, the latest applications concerning the use of microwave energy for the determination of hydrocarbon contaminants—namely polycyclic aromatic hydrocarbons (PAHs) and mineral oil hydrocarbons (MOH)—are reported and critically overviewed and future trends are delineated.

## 1. Introduction

The first application of microwave energy dates back to 1986, when Ganzler et al. [1], investigated the applicability of microwave irradiation to the extraction of various types of compounds from soil, seeds, foods and feeds, prior to chromatographic determination. Since then, numerous publications concerning the use of microwaves for the extraction of fats, bioactive and nutrient components, inorganic (metals) and organic contaminants, from plant material and foods, have been published. 

Concerning organic contaminants, microwave-based techniques have been successfully applied to pesticides, polychlorinated biphenyls, *N*-nitrosamines, mycotoxins, residues of veterinary drugs and last but not least, hydrocarbon contaminants.

The latter comprise well known food contaminants such as polycyclic aromatic hydrocarbons (PAHs), as well as emerging contaminants such as mineral oil hydrocarbons (MOH), which have attracted great attention in the last decade.

PAHs are a large class of organic compounds containing 2–6 fused benzene rings with no alkyl substituents, generated by incomplete combustion of organic matter. They are both environmental and processing contaminants. Environmental contamination is mainly due to natural (e.g., forest fires, volcanoes, etc.) and anthropogenic (mainly the combustion of fossil fuels) sources. Food processing involving thermal treatments at high temperature and/or direct contact with combustion gases (smoking, toasting, roasting or grilling) is responsible for the high PAH levels found in processed foodstuffs [2]. As early as 1976, the United States Environmental Protection Agency (US EPA) selected “16 US EPA priority PAHs” to assess the health risks due to environmental exposure. Since then, these 16 PAHs have been widely accepted as reference PAHs to be monitored in food. Based on new toxicological evidences, Recommendation 2005/108/EC [3] introduced a different list of “15 + 1 EU priority PAHs” to be monitored in food (8 of which had never been researched before). In 2011, the EC enacted Regulation 835/2011, fixing limits for benzo (a) pyrene (BaP) and the sum of 4 PAHs (PAH4) for several food classes. Moreover, Regulation 836/2011 [4] fixed performance criteria for their analytical determination which is mainly conducted via high performance liquid chromatography (HPLC), coupled to spectrofluorometric detection (FLD) or by gas chromatography-mass spectrometry (GC-MS) (SIM mode). 

MOH comprise thousands of hydrocarbons (isomers) of petrogenic origin, which, contrarily to PAHs, cannot be separated into single peaks but can be fractionated into two classes of different toxicological relevance—the mineral oil saturated hydrocarbons (MOSH) and the mineral oil aromatic hydrocarbons (MOAH), the latter mainly composed by 1–3 ring alkylated compounds. MOSH are of concern because of their bioaccumulation potential in human tissues, while MOAH (particularly those with 3 or more rings) may comprise suspected carcinogens (including little amounts of PAHs) and for this reason their presence in food should be avoided or maintained “as low as reasonably achievable” [5]. MOH can reach the food through different routes (environment, field production, food processing and packaging). Since packaging made of recycled cardboard may contain huge amounts of MOH from different sources (printing ink, adhesives, solvents, waxes and additives used in the production process), great concern has been raised over recent years about their possible migration into food. At present there is no approved legislation regulating the presence of mineral oils in food and packaging.

The reference method for MOSH and MOAH quantification in food is the on-line HP(LC)-GC-FID (high performance liquid chromatography-gas-chromatography-flame ionization detection) method developed by Grob and collaborators [6,7], also approved as an ISO (International Standard Organization) method for vegetable oils and foodstuffs on the basis of vegetable oils (BS EN 16995: 2017).

The analytical determination of both PAHs and MOH must be preceded by an extraction step, whose importance is often underestimated and which should guarantee quantitative recovery with minimal solvent consumption [2]. In this context, extraction carried out with traditional methods (Soxhlet extraction, liquid-liquid or solid-liquid extraction, traditional saponification, etc.) involves large volumes of solvent and intensive sample manipulation, it is time consuming and can lead to contamination during sample preparation and poor reproducibility. 

Over the last years, microwave-assisted extraction (MAE)—also called microwave-assisted solvent extraction (MASE)—has become a viable alternative to the conventional techniques due to substantial advantages in sample preparation, as it requires a much lower solvent volume, reduces extraction time and allows the processing of a large number of samples simultaneously. Several comparative studies have shown the possibility with MAE to obtain excellent performance in terms of recoveries and precision, compared to other traditional extraction techniques. 

Basically, microwave-assisted extraction can be performed by using closed or open vessel giving rise to different extraction modes, namely pressurized microwave-assisted extraction (PMAE), when using closed vessels and atmospheric pressure microwave-assisted extraction (APMAE), also known as focused microwave-assisted extraction (FMAE), when using open vessels. 

Even though many applications report the use of domestic microwave systems, particularly when using closed vessels under pressure, this practice is not recommended from a safety point of view. Nowadays, dedicated apparatus, able to carry out MAE in total safety, are available on the market. When using PMAE, temperatures well above the boiling point of the solvent can be reached, by significantly increasing analyte desorption rate, thus speeding up the process. In case of APMAE, maximum temperature coincides with the solvent boiling point. FMAE systems are generally used in this case. 

The main aim of this review is to give an updated overview of microwave-based extraction techniques by describing the principles of the techniques, instrumentation, extraction mode, parameters affecting the extraction, integrated techniques for further purification/enrichment. Latest applications concerning hydrocarbon contaminants and future trends will also be illustrated and discussed. With some exceptions for older significant works, scientific papers published over the last 10 years were considered for this review (Web of Science, Scopus, Pubmed, until 30 August 2019). 

## 2. Principle of Microwave Heating

Microwaves are non-ionizing radiations with a range of frequencies between 300 and 300,000 MHz, able to activate the rotational energy levels of the molecules. The microwaves used for scientific and domestic purposes have a frequency of 2450 MHz.

The principle of microwave heating is based on the effects of dipole rotation and ionic conduction that microwaves exert on dipolar and charged molecules (ions).

Dipolar molecules, are sensitive to the alternating electric field generated by microwaves, which, continually changing its direction, causes the molecules to rotate quickly (4.9 × 10^9^ times per second) to align their own dipole with that of the electric field. Ionic conduction refers to the electrophoretic migration of ions in a solution under an electromagnetic field. The friction between molecules/ions and the solution causes a rapid heating [8,9].

Different from traditional heating sources, which require some time to heat the core of the sample (heat transfer occurs by conduction and/or convection), microwaves act on the whole sample volume (if the medium is homogeneous) or on localized heating centres constituted by polar molecules contained in the product, thus allowing for rapid heating, maintaining the temperature gradient low. 

## 3. Instrumentation

The main components of a microwave system include a microwaves generator (magnetron), an insulator (which has the function to protect the magnetron from reflected energy not absorbed by the sample), a waveguide for energy transmission, a resonance cavity and a power supply [10]. 

PMAE apparatus working with closed vessels under pressure (Figure 1a), are by far the most utilized. They have interchangeable rotating carousel of different number of maximum positions for simultaneous multiple extraction. A mode stirrer is positioned on the upper wall of the cavity to ensure a better distribution of the microwave energy. Almost all microwave extraction systems are equipped with a magnetic stirring system to favour surface contact between sample and solvent, improving extraction efficiency and reducing solvent consumption. The rotating magnet is placed under the floor of the cavity, while the magnetic anchors are placed directly in the extraction cell. Extraction cells consist of inert materials (polytetrafluoroethylene, perfluoroalkoxy, glass or quartz) transparent to microwaves, sometimes inserted in containers mechanically resistant to high pressures, such as polyetherimides or polyetheretherketone (PEEK) reinforced with glass fibre. 

While first generation extraction cells could work up to a maximum of 7–14 atm, nowadays extraction cells are available for applications over 100 atm and maximum operating temperatures up to 200–300 °C. Extraction cells designed to work at high pressure are equipped with different over-pressure release systems, useful for avoiding the risk of explosion if the maximum sealing pressure is exceeded. A pilot cell is dedicated to temperature and pressure control. Temperature control is necessary to optimize extraction efficiency, prevent thermal degradation and obtain reproducible performances. A feedback control system makes it possible to modulate the magnetron power as a function of the set temperature, avoiding sample over-heating. Many systems have an external infrared sensor on the side wall or below the floor of the cavity to monitor the temperature in each vessel. An internal pressure sensor monitors the pressure in the pilot cell, while an external sensor continuously monitors the quality of the air expelled from the system, blocking the power supply to the magnetron, if necessary. 

FMAE apparatus resemble a Soxhlet extractor and have adjustable power from 0–300 W—the sample is weighed in a cellulose thimble and inserted into the quartz extraction cell containing the solvent. The particular design of the waveguide enables microwaves focusing on the lower part of the container where the sample-solvent mixture is placed. A reflux condenser is mounted over the extraction cell to allow for solvent re-condensation, without the need to control temperature and pressure. To optimize the extraction process, it is possible to adjust extraction time, microwave power, nature and volume of the solvent.

Figure 1b shows the first apparatus commercially available (Soxwave 100, Prolabo Ltd., Fontenaysous-Bois, France), which is out of production since 2000, later replaced with multi-position systems able to process 2–6 samples simultaneously.

The most important advantage of this system is that it allows us to obtain a more homogeneous and reproducible treatment of the sample and reduces the dispersion of electromagnetic energy, thus allowing us to decrease the powers applied to obtain a specific effect (even 10 times). In addition, the equipment is less expensive) as it is not necessary to use containers compatible with high pressures and it is particularly suited for thermolabile analytes. Finally, once the process is complete, it is not necessary to wait for sample cooling before opening the extraction cell. 

## 4. Extraction Mode

Depending on the nature of the analyte and matrix, solvent extraction can take place according to different modes [8].

For polar or medium polar analytes, the sample is usually immersed in a solvent or in a mixture of solvents that strongly absorb the microwave energy and well solubilize the analytes.

In case of extraction of thermolabile analytes and essential oils from vegetable matrices with a high water content, the sample can be extracted with a solvent transparent to microwaves (i.e., hexane). The microwaves interact with the free water present in the plant tissue, causing rapid expansion with consequent cell rupture. In this way the essential oil or analytes of interest flow towards the organic solvent which remains cold (or heats up weakly).

Apolar organic compounds such as hydrocarbon contaminants, which are of major interest for this review, are well soluble in non-polar solvents transparent to microwaves. Nevertheless, the sample can be suitably extracted with a combination of 2 solvents (i.e., hexane/acetone mixtures), the first one apolar, transparent to microwaves but able to extract the analyte and the second one, polar (high dielectric constant), able to rapidly heat the sample-solvent mixture. Of course, such combination has the disadvantage of favouring the extraction of compounds of a wide range of polarity [8].

As an alternative, to achieve more selective extraction, the sample, previously dehydrated, is immersed in a microwave-transparent apolar solvent (for example hexane) in the presence of a secondary microwave absorber (a chemically inert fluoropolymer filled with carbon) able to absorb the microwave energy and quickly transfer the thermal energy generated to the non-polar solvent. Carboflon bars (CEM) have been, for example, used for extracting PAHs from freeze-dried smoked meat using hexane as a solvent, obtaining cleaner extracts than those obtained with a mixture of hexane/acetone, which gave comparable results but less selective extraction [11].

## 5. Sample Processing

Before extraction, the sample has to be homogenized in order to guarantee a representative sampling. Solid samples need to be finely grounded to improve the sample-solvent contact. Liquid samples can be also processed as they are or adsorbed on a solid matrix. 

When the presence of water in the sample has a negative impact on the extraction yields (i.e., in case of apolar analytes to be extracted with an apolar solvent) and/or reproducibility (different water content can lead to different sample heating), the sample needs to be dehydrated or lyophilized or the water content has to be standardized in order to obtain uniform heating. 

The ground sample (1–5 g) is directly weighed into the extraction vessel, added with the solvent (25–30 mL) and the extraction cell is sealed and positioned into the microwave oven.

When using PMAE, a pre-heating step precedes sample extraction. The time required to reach the working temperature will depend on the number and type of samples but generally takes less than 2 minutes. Extraction generally takes 10–20 min but before opening the cells after extraction, sample extracts have to be cooled at room temperature to avoid volatile losses. 

The final extract is separated from the solid particles by decantation, centrifugation or filtration and subjected to analytical determination with the usual methods. As reported in Section 7, in most cases it is necessary to proceed with a further purification step to eliminate any co-extracted interference.

## 6. Parameters Affecting the Extraction

The choice of the extraction solvent represents one of the most important parameters to be optimized. When choosing the solvent, it is important to consider its ability to absorb microwaves, disrupt interactions with the matrix, solubilize the analyte but also its selectivity towards the analyte of interest [8]. Non-polar solvents, which are well suited for the extraction of apolar contaminants, are transparent to microwave and, as already explained, require the use of secondary absorber or solvent mixtures of high and low dielectric constant, enlarging the range of co-extracted compounds.

The solvent volume must be sufficient to ensure that the whole sample is immersed. In general, the amount of solvent needed for a sample may be about 10-fold less than the amount required for classical extraction.

In general, with the exception of thermolabile analytes, extraction efficiency increases by increasing the temperature. In some cases, lower recovery was observed at higher temperature, mainly due to reactions taking place in the presence of the matrix [8]. The optimal extraction temperature depends not only on the analyte but also on the matrix and needs to be determined experimentally [12]. Extraction times are considerably shortened compared to traditional techniques; often times of 5–10 min are sufficient, against times of the order of the hours, necessary with the traditional extraction methods. In some cases (for example extraction of pesticides or some aromatic amines) use of long extraction times worsen the recoveries, while in other cases they do not affect recoveries [12].

When developing a method, it is always useful to test the effect of temperature and extraction time. Sometimes, even though optimal recoveries can be obtained at higher temperatures and times, it is more convenient to use lower extraction temperatures or shorter times in order to minimize co-extraction of interfering compounds.

## 7. Combined Extraction/Purification Techniques Applied to Food Contaminants

MAE has been recognized as a powerful technique for isolating organic contaminants of different polarity from food. However, its high extraction efficiency is responsible of co-extraction of food matrix components which could give interference during quantification. To overcome this problem, different protocols based on the combination of MAE with other pre-treatments aimed at improving sample purification have been developed over the years. 

Solid phase extraction (SPE) and/or liquid-liquid extraction (LLE), have been successfully used for long years but they still make use of organic solvent.

When high amount of fat is co-extracted with the target analytes, saponification can be advantageously applied to eliminate triglycerides. Saponification carried out with traditional methods involves reflux of the sample at boiling temperature in the presence of alcoholic KOH, followed by several extraction steps of the unsaponifiable with an organic solvent and washing with water. Microwave assisted saponification (MAS) combined with simultaneous unsaponifiable extraction has proved to be a powerful alternative to traditional saponification. It enables to achieve rapid, efficient extraction and simultaneous purification and enrichment of apolar contaminants in fatty food, using reduced volumes of organic solvent and saving time [13,14]. 

Recently, some authors [15] have reviewed green procedures combining MAE with different micro-extraction techniques, in order to drastically reduce (or eliminate) the use of organic solvents. Most of these applications regard food contaminants. 

Among these, it is worthy of mention the combination of MAE with dispersive solid phase micro-extraction (D-µ-SPE), till now not applied to hydrocarbon contaminants but successfully applied to the extraction of pesticides [16,17], antibiotic residues [18] and mycotoxins [19] from a wide range of food. The combination of MAE (more precisely MAS, because a basic hydrolysis is performed) with dispersive liquid-liquid micro-extraction (DLLME) is for sure the most utilized method, for extraction/purification of food contaminants, particularly of hydrocarbon contaminants (as detailed in the next paragraphs). Such a combination has been successfully applied also to the determination of residues of veterinary drugs from fish [20] and N-nitrosamines from meat products [21].

D-µ-SPE, recently reviewed by Chisvert et al. [22] is usually employed for matrix clean-up purposes, that means after the dispersion of sorbents any possible matrix interferences is retained on the sorbents. Then sorbents are separated from the bulk solution by centrifugation and the target analytes are recovered in the supernatant. On the contrary, D-µ-SPE can be also performed by trapping the target analytes in the sorbents. After separating the sorbent from the matrix solution by centrifugation, filtration or a magnetic field (when using magnetic nanoparticles as sorbent), the target analytes are eluted or desorbed by an appropriate solvent. Nature and physicochemical properties of the solid sorbent are very important in order to achieve sensitive and selective determination of target compounds. D-µ-SPE has been recently applied for PAH isolation from water sample [23], but, as already mentioned, not in combination with MAE.

DLLME, first introduced by Rezaee et al. [24], is based on a ternary component solvent system (sample solution, dispersive solvent and extraction solvent). An appropriate mixture of an extraction solvent (high-density organic solvent) and dispersive solvent (water-organic miscible solvent), is rapidly injected into the aqueous sample by a syringe, resulting in the formation of a cloudy solution containing fine droplets of the extraction solvent dispersed entirely in the aqueous phase. After centrifugation, the analytes are separated and enriched into the organic phase (extraction solvent) prior to GC or HPLC analysis. Figure 2 shows a schematic representation of DLLME steps.

## 8. Applications Focusing on Hydrocarbon Contaminants

### 8.1. Focused Microwave-Assisted Extraction (FMAE)

Applications involving the use of FMAE (or APMAE) for extracting hydrocarbon contaminants from food matrices, are very limited.

Soclo et al. [25] combined a two-step FMAE for PAH extraction from biota (comprised mussels and fish tissue). Hydrocarbon contaminants were firstly co-extracted with the fat (using dichloromethane as MAE solvent), followed by a microwave treatment with concentrated sulphuric acid (added to the cooled CH_2_Cl_2_ extract directly into the same FMAE tube) to destroy lipids, letting the PAHs unaffected. After repeated washing with water, the final dichloromethane extract was purified on an alumina micro-column and fractionated on a silica micro-column, first with pentane to collect saturated hydrocarbons and then with a mixture of pentane/dichloromethane (65:35, *v*/*v*) to recover the aromatic fraction. The purified aromatic fraction was further separated into three fractions by NP (normal-phase)-HPLC, concentrated and analysed by GC-MS.

Even though the great majority of the published works on PAHs are based on their chromatographic determination (via HPLC or GC), simpler and rapid non-chromatographic method based on a combination of MAE and fluorescence spectroscopy have been also proposed in recent years.

Li et al. [26] developed a rapid and cheap method combining APMAE pre-treatment (using a domestic microwave oven), with a novel synchronous fluorescence spectroscopic approach for rapid detection of three PAHs in tea samples. The dried tea sample (0.5–1.0 g) was placed in an uncovered conical flask and extracted with dimethyl sulfoxide (140 W for 4 min). Meanwhile, a cup of 400 mL of water was placed in the microwave oven to avoid sample overheating. After cooling, the extract was filtered, added with 2% sodium sulphate solution and extracted with hexane. Nor further purification or chromatographic separation was required before fluorometric detection. 

By using advanced chemometric tools to resolve partially overlapping fluorescence spectra, Alarcón et al. [27,28] elaborated a modern approach to improve the selectivity of fluorescence measurements, thus avoiding chromatographic separation. Under optimized conditions, 1 g of the edible oil was weighed into a 50 mL Erlenmeyer flask, added with acetonitrile (30 mL) and connected to an air-cooled condenser. The glass system was placed in the microwave oven and heated for 19 min at 150 W. Since matrix interference did not allow for direct PAH detection after FMAE, the effectiveness of different SPE sorbents (silica, C18 and graphitized carbon black) in removing interference by tocopherols and pigments (chlorophyll and pheophytin), was examined. The best results were obtained when using silica cartridges. 

As detailed in a next paragraph, Germán-Hernández et al. [29] proposed an interesting FMAE method using microwave-assisted micellar extraction (MAME) and aggregates of ionic liquid (IL) as surfactant, for PAH extraction from toasted cereals.

### 8.2. Pressurized Microwave-Assisted Extraction (PMAE) 

PMAE, followed by RP (reversed phase)-HPLC and spectrofluorometric detection, has been successfully used by different authors for PAH extraction and determination in various foods, such as edible oils, bakery products, fish and meat. Depending on the matrix, more or less intensive sample purification steps were required before analytical determination. 

Hernández-Póveda et al. [30] used a conventional microwave oven and hermetically closed PTFE (polytetrafluoroethylene) reactors for extraction of 15 PAHs from cookies. Optimal extraction yields were obtained when extracting 5 g of sample with 10 mL of acetone/hexane (1:1, *v*/*v*) for 9 min at 700 W. Co-extracted fat was removed by liquid-liquid extraction with acetonitrile, followed by saponification. A further SPE clean-up was required prior to HPLC-FLD analysis. Compared to conventional Soxhlet extraction and ultrasound assisted extraction, the optimized MAE procedure was faster (9 min) and solvent consumption was reduced to only 25 mL per sample. Recoveries ranged from 96 to 105%, while LOD (limit of detection) were comprised between 0.015 to 0.7 µg/kg.

A simpler PMAE method followed by a rapid SPE clean-up was developed by Purcaro et al. [11] for PAH determination in smoked meat. The authors compared the performance obtained with the mixture hexane/acetone (the most utilized when analysing environmental matrices), with that obtained with hexane alone. Both solvents gave quantitative extractions but if on the one hand the presence of acetone allowed for microwave heating, on the other it led to co-extraction of a wider range of interfering compounds, so the latter was chosen. Following optimized conditions, 2 g of lyophilized sample added with 20 mL of *n*-hexane and a Carboflon bar (a strong microwave absorber made of carbon black encapsulated in a glass envelope) were heated at 115 °C for 15 min. Prior to RP-HPLC-FLD analysis, the sample extract underwent a clean-up step on a silica cartridge according to Moret and Conte [31]. Compared to solvent extraction assisted by sonication, PMAE allowed to obtain better extraction efficiencies. Recovery of heavy PAHs ranged from 83 to 103%, while LOD were lower than 0.2 µg/kg for all heavy PAHs, except for IP (0.4 µg/kg).

A procedure involving preliminary hydrolysis with KOH-ethanol/water, 9:1 *v*/*v* (in a water bath at 60 °C) followed by MAE extraction of the hydrolysed sample with 10 mL of cyclohexane (10 min at 100 °C) and SPE purification on neutral Al_2_O_3_ prior to HPLC-FLD determination, was developed and applied by Zhang et al. [32] to isolate PAHs from aquatic products. 

Ramalhosa et al. [33] developed and validated a sensitive method for determining 18 PAHs in fish samples. Analysis was performed by PMAE followed by HPLC analysis with photodiode array detector (DAD) and FLD. Response surface methodology was used to find out the optimal extraction conditions which consisted in extracting, under medium stirring speed, 0.5 g of homogenized sample with 10 mL of acetonitrile for 20 min at 110 °C. Different from previously published methods, clean-up of MAE extracts was not necessary due to the selectivity of the solvent chosen. With respect to the mixture hexane/acetone (1:1, *v*/*v*) recommended by the US Environmental Protection Agency (EPA method 3546) for the extraction of different organic pollutants from environmental samples, acetonitrile has the advantage of high dielectric constant in addition to be compatible with the HPLC-FLD procedure (no solvent exchange was required, thus reducing possibilities of analyte losses during sample preparation).

Gharbi et al. [34] used PMAE for investigating the presence of PAHs and MOH in olives. Extraction conditions were 5 g of olive paste, added to 20 mL of hexane/ethanol 1:1 (*v*/*v*), extracted at 120 °C for 20 min. After cooling, the extract was added to water directly into the extraction vessel, to separate the hexane from the ethanol/water phase. To isolate PAHs from triglycerides, SPE on a 5 g silica cartridge according to Moret and Conte [31], was applied, followed by HPLC-FLD analysis. For MOH analysis, an aliquot of the extracted fat was diluted and injected directly into the on-line HPLC-GC system. Epoxidation was performed when required to eliminate interference by olefins [6].

### 8.3. Microwave-Assisted Saponification (MAS)

Since MAS is usually performed in closed vessels under pressure, it can be considered a variant of PMAE. Starting from 2000, MAS has been widely used to speed-up extraction/purification of hydrocarbon contaminants from various foods (Table 1). Its application to other contaminants regards the determination of PCBs (polychlorinated biphenyls) in mussels [35], organochlorine pesticides in oyster samples [36] and polybrominated flame retardants in aquaculture samples [37]. The first MAS applications for PAH determination in food involved a microwave assisted saponification followed by traditional liquid-liquid extraction [38,39].

Table 1, summarizes MAS applications on hydrocarbon contaminants.

For determining 16 PAHs in pumpkin seed oil, Gfrerer and Lanmayer [39] used a commercial microwave extractor. Optimized conditions are reported in Table 1. The authors found that KOH concentration (2.5–5 M) higher than the optimal one (1.5 M) caused gel-like formation, preventing liquid-liquid extraction and determining emulsion formation. Unsaponifiable extraction was performed traditionally after transferring the extract into a separatory funnel. An SPE on a mixed bed of activated silica- and cyano-phase was required before GC-MS analysis.

Hernández-Borges et al. [40] used MAS for determining 19 aliphatic hydrocarbons and 27 PAHs (including some alkylated ones) in marine biota (mussels). Microwave-assisted saponification and unsaponifiable extraction were performed in succession. Experimental design (ED) and artificial neural networks (ANNs) were used to optimize the experimental conditions. NIST-CRM 2978 (certified reference material) was used to test the validity of the developed method which showed a good agreement with certified values.

Different to previous authors, Pena et al. [41] optimized and used, for the first time, a one-step microwave-assisted procedure (for PAH determination in fish), involving saponification with KOH in methanol and simultaneous LLE extraction of PAHs with *n*-hexane (directly added into the extraction vessel). Experimental design methodology allowed for quick and robust optimization of extraction parameters. Optimized conditions are reported in Table 1. Since fish samples having different fat content need different amount of alkali to warrant complete hydrolysis, an excess of alkali suitable for all types of samples tested, was chosen. A further clean-up on silica gel involving low solvent consumption (a few millilitres) was required to eliminate interferences. Recovery around 90% and quantification limits far below the regulated limits, were obtained. Accuracy validation was carried out using NIST SRM (standard reference material) 2977.

Navarro et al. [42] optimized PAH extraction from biota samples (oysters, mussels and fish liver) and compared MAS (without further clean-up) with MAE (extraction with acetone) followed by SPE clean-up (on Florisil or silica cartridges) or gel permeation chromatography (GPC) prior to GC-MS. MAE followed by GPC gave the cleanest extracts. Also, SPE on 5 g Florisil cartridges provided good results. In addition, the concentrations obtained for NIST SRM 2977 (mussel) were in good agreement with the certified values. In the case of MAS (directly followed by GC-MS), less clean extracts were obtained, leading to overestimation of the heaviest PAH concentrations. 

Based on the work of Pena et al. [41], Akpambang et al. [13] optimized microwave-assisted saponification/extraction followed by SPE on silica gel and HPLC-FLD analysis, for PAH determination in smoked meat and fish from the Nigerian market. The same procedure was later applied to roasted plants usually eaten as snack in Nigeria [44] and with little modifications for PAH determination in propolis and propolis-based extracts [43]. In the latter case no further purification was required before HPLC-FLD determination. Preliminary tests for PAH extraction from raw propolis were carried out by using *n*-hexane as extraction solvent. Unfortunately, a large amount of co-extracted waxes remained after solvent removal (cause of low recoveries and poor repeatability). To avoid this problem, a saponification step was introduced in order to hydrolyse the waxes. As visible in Figure 3, which reports the HPLC trace of a highly contaminated raw propolis (1372 µg/kg of BaP) after simultaneous saponification/extraction, the sample is free from interferences and can be directly injected into the HPLC apparatus. About half of the samples analysed presented BaP concentrations exceeding 2 μg/kg, which is proposed as a regulatory limit for dietary supplements. 

Moret et al. [14] developed and validated a MAS method for extraction/enrichment of MOH from cereal-based products (bread, pasta, biscuits and bakery products). According to the method, an aliquot of the ground sample (5 g) is directly weighed into the extraction cell, added to the internal standards, 10 mL of a saturated KOH solution in methanol, 10 mL of *n*-hexane and saponified in the microwave extractor for 20 min at 120 °C. After cooling, the extract is rapidly washed with water and methanol (into the extraction cell and then injected into the on-line LC-GC instrument, directly or after concentration. With respect to the method proposed for exhaustive MOH extraction from some dry food [45], involving soaking with hot water followed by a double extraction first with ethanol and then with hexane overnight), the validated procedure was more rapid, reduced solvent consumption and allowed for simultaneous enrichment. Later, the same procedure was successfully applied to a wide range of other food products such as milk, milk powder, baby foods, coffee, fish, and, with some modifications, to edible oils. 

In conclusion, MAS can be applied, without any pre-treatment, to all types of food (dry and wet, with a low or high fat content) and has the advantage to effectively remove high amounts of fat and other interferences allowing for simultaneous enrichment and purification. Figure 4 shows the HPLC-GC trace of a biscuits sample before and after concentration following MAS.

### 8.4. MAE Combined with DLLME

As emerges from previous examples, MAE has been employed in combination with conventional LLE and/or SPE and microwave-assisted hydrolysis/saponification (MAS) has proved to be an advantageous tool for simultaneous extraction/purification.

Starting from 2012, most of the published papers focused on the combination of MAE (MAS) and DLLME, going towards further simplification and miniaturization of the sample preparation. Table 2 summarizes the latest applications regarding this combination for detecting hydrocarbon contaminants.

A simple and efficient method using MAE combined with DLLME and GC-MS analysis was developed by Ghasemzadeh-Mohammadi et al. [46], for the extraction and quantification of 16 PAHs in smoked fish. BaP, chrysene (Ch) and pyrene (P) were employed as model compounds and spiked to smoked fish to assess the extraction procedure. Several parameters, including the nature and volume of hydrolysis solution, extracting and disperser solvents, microwave time and pH, were optimized. Under optimized condition 1 g of fish sample underwent hydrolysis/extraction with 12 mL of a KOH 2 M/ethanol (50:50 *v*/*v*) mixture, in a closed-vessel system. Ethanol was chosen as the organic modifier because it provided cleaner chromatograms with respect to methanol. After cooling, the sample was centrifuged and the aqueous phase was transferred into another vessel and pH was adjusted to 6.5 by adding hydrochloric acid. The pH of the sample solution before DLLME plays an important role. In fact, if it is too high, no sedimented phase is formed at the bottom of the vessel after centrifuging. For DLLME, 500 μL of acetone (disperser solvent) containing 100 μL of ethylene tetrachloride (extraction solvent) was rapidly injected by syringe into 12 mL of the sample extract solution, thereby forming a cloudy solution. Phase separation was achieved by centrifugation and a volume of 1.5 μL of the sedimented phase was analysed by GC-MS in selected ion monitoring (SIM) mode. The MAE-DLLME method coupled with GC-MS provided excellent enrichment factors (in the range of 244-373 for 16 PAHs), good repeatability (with a relative standard deviation between 2.8 and 9%) and recoveries higher than 80%.

Later, the same authors [47] used surface response methodology to better optimize the extraction condition (Table 2), studying interactions among different parameters. After pH adjustment, before DLLME, the sample was added with Carrez solutions I (potassium hexaferrocyanide) and II (zinc acetate) to precipitate protein and soluble carbohydrates. All of the 80 smoked fish samples, except three, had BaP concentration below the European Commission’s maximum level of 2 μg/kg for smoked fish, while PAH4 varied between 3 and 12 μg/kg wet weight. 

Similar procedures, with little modifications, were proposed for PAHs extraction and determination in grilled meat [48], smoked rice [49], bread [50,51], coffee [52], vegetables [53] and edible oils [54]. 

In case of edible oils [54], PAHs were first extracted with acetonitrile/acetone (50:50, *v*/*v*) and then saponified with methanolic KOH, in two separate steps. Then a mixture of ethanol (disperser solvent), tetrachloroethylene (extraction solvent) and biphenyl (internal standard) was rapidly injected into the sample solution. After phase separation, the sedimented phase was analysed by GC-MS. 

Figure 5 shows GC-MS chromatograms of toasted bread sample [51] analysed with the proposed method, which clearly indicated no significant matrix effect. 

As can be inferred from Table 2, chlorinated solvents (ethylene tetrachloride or chloroform) are generally selected as an extraction solvent. In general, the extraction solvent should demonstrate good chromatographic behaviour (elution strength lower than the mobile phase used in the separation system), high extraction efficiency for the target analytes and should form very tiny droplets (cloudy solution) in the presence of a disperser solvent when injected into an aqueous solution. Furthermore, in conventional DLLME, the extraction solvent should be denser than water, have a low solubility in water and be miscible with the dispersive solvent. 

To overcome the disadvantage of consuming high-toxicity chlorinated solvents, low toxic alcohol and brominated solvents were also tested as extraction solvents for PAH extraction from vegetables [53] and 1-bromo-3-methylbutane was finally chosen. 

The selection of disperser solvent is also critical in DLLME. As a disperser, acetone or acetonitrile are generally used (the first one in combination with ethylene tetrachloride as extraction solvent and the second one in combination with chloroform). The miscibility of the disperser solvent in the aqueous sample solution and in the organic extraction solvent is the most important parameter for its selection, meanwhile extraction time has poor impact on the extraction efficiency. Indeed, extraction and the equilibrium state is accomplished very rapidly after formation of the cloudy solution. Similarly, centrifugation time has little effect on extraction yield. Due to the large contact surface between the tiny drops of extraction solvent and the sample, the equilibrium is achieved in a few seconds. Centrifugation helps the cloudy solution to settle to the bottom of the tube. A centrifugation time of 5 min at 4000 rpm is generally chosen.

### 8.5. Ionic Liquids (ILs)

Ionic liquids (ILs) are non-molecular solvents which over the last years have gained significant attention as alternative extraction solvents. The term ILs is used to describe salts in liquid form, having melting points virtually close or below normal room temperature. Basically, due to their attractive physiochemical properties in the extraction process, which is influenced by the characteristic features of their cations and anions, they are considered optimal substitutes for traditional organic solvents. They are easily miscible with other solvents, non-inflammable, have low chemical reactivity, good thermal stability and minimal vapor pressure [55]. They effectively absorb microwave radiation and dissipate energy swiftly through ionic conduction [56]. 

Up till now use of ILs in combination with MAE (followed by D-µ-SPE or DLLME), has been only described for the extraction of pesticides from food [57,58,59]. An interesting example regards the use of poly(ionic liquids) immobilized on magnetic nanoparticles, used to extract pesticides from fruits and vegetable samples [60].

Since ILs with long aliphatic substituents may undergo micellization in aqueous solutions, ILs aggregates have been also proposed as an alternative to traditional surfactant systems in microwave-assisted micellar extraction. Extraction of organic contaminants from solid matrix by means of micellar media of traditional surfactant systems or ionic liquid aggregates can be accelerated by means of microwaves or ultrasound. The former has the advantage of requiring lower concentrations of the extracting agents, which results very convenient from an environmental point of view. 

Germán-Hernández et al. [29], optimized a FMAE method using aggregates of the ionic liquid (IL) 1-hexadecyl-3-butylimidazolium bromide (HDBIm-Br), followed by HPLC-UV/FLD determination of PAHs in toasted cereals.

The optimized extraction method which employed low amounts of sample (0.1 g) and IL reagent (77 mg), avoided completely the use of organic solvents and allowed to reach, for 15 EU-priority PAHs, LOD between 0.02 and 4.0 ng/mL, recoveries from 70 to 109% and precision values lower than 13% (as relative standard deviation). The optimum microwave extraction conditions were as follows—microwave power of 50 W to reach a maximum temperature of 80 °C in 4 min and a hold time of 10 min. After cooling, the extract was centrifuged and the supernatant injected without any further clean-up.

To reach higher sensitivity, soon later the same research group [61] described an improved method using modified IL-based surfactants. LOD lower than 0.03 μg/kg and recovery comprised between 80–95%, were obtained. 

## 9. Conclusions and Future Perspectives 

Extraction, when carried out with traditional methods, often represents the bottleneck of the entire procedure. With respect to conventional extraction technique, MAE allows the performance of efficient and rapid extraction, with reduced solvent consumption. Like the majority of other organic food contaminants, PAHs and MOH are non-thermolabile, therefore, to reach high extraction rate in a short time, extraction can be advantageously carried out at temperature higher than the solvent boiling point, using PMAE, whose use prevails by far over FMAE.

Depending also on the complexity of the food and the selectivity of the solvent used for the extraction, PMAE can lead to co-extraction of huge amounts of interfering compounds, thus making further purification necessary. With respect to MAE, MAS represents an enhanced microwave-based technique, enabling for simultaneous extraction and purification/enrichment. It has been successfully applied to extract and concentrate PAHs and MOH from a wide range of foods. 

Over the last decade, sample preparation protocols combining MAE (or MAS) with micro-extraction techniques, such as D-µ-SPE and DLLME have found increasing applications for a wide range of food contaminants. With respect to conventional LLE and SPE, DLLME provides several advantages such as simplicity of operation, rapidity, low cost and increased sensitivity. It also allows us to obtain high recovery and good repeatability. ILs, as green extraction solvents, may represent a suitable alternative to common toxic solvents. However, commercially available ILs are relatively expensive and not compatible with GC analysis. Therefore, further work should focus on the development of new extraction phases with low toxicity and good compatibility with both GC and HPLC analysis.

## Figures and Tables

**Figure 1 foods-08-00503-f001:**
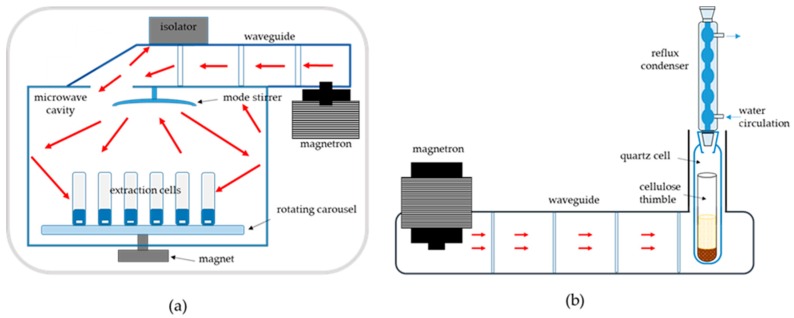
(**a**) Diffused microwave extractor for pressurized microwave-assisted extraction (PMAE); (**b**) Focused microwave extractor for atmospheric pressure microwave-assisted extraction (APMAE).

**Figure 2 foods-08-00503-f002:**
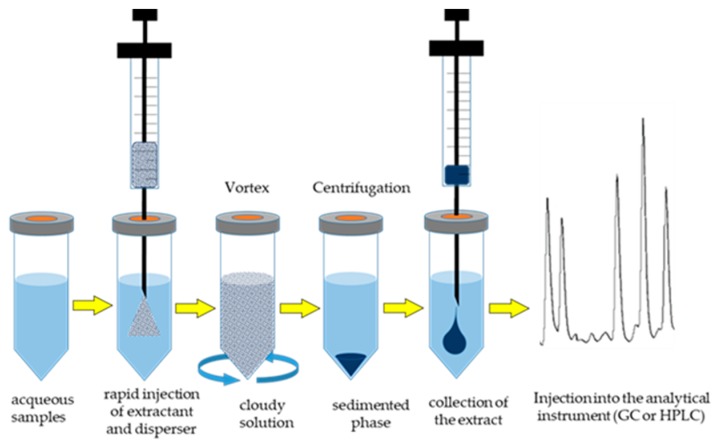
Schematic representation of dispersive liquid-liquid micro-extraction (DLLME).

**Figure 3 foods-08-00503-f003:**
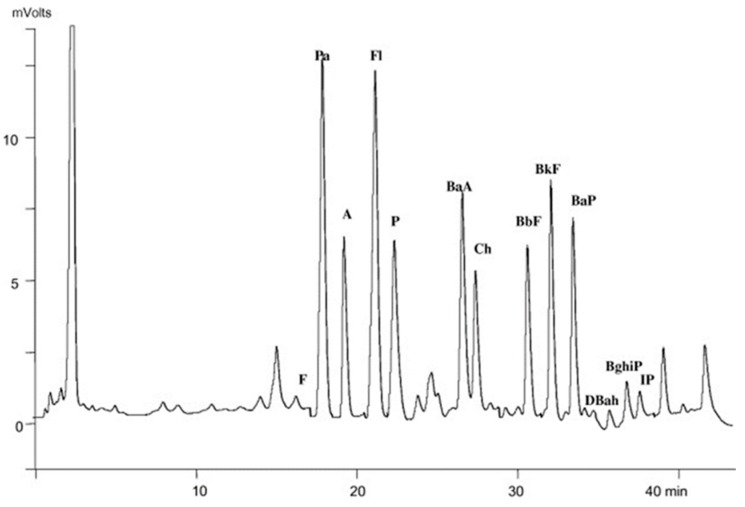
HPLC-FLD (high performance liquid chromatography-spectrofluorometric detection) trace of a sample of raw propolis analyzed directly after MAS, (F) fluorene, (Pa) phenanthrene, (A) anthracene, (Fl) fluoranthene, (P) pyrene, (BaA) benz[a]anthracene, (Ch) chrysene, (BbF) benzo (b) fluoranthene, (BkF) benzo[k]fluoranthene, (BaP) benzo (a) pyrene, (DBahA) dibenz[a,h]anthracene, (BghiP) benzo[g,h,i]perylene, (IP) indeno (1,2,3-*cd*) pyrene. Reprinted from Ref [43] with permission from Elsevier.

**Figure 4 foods-08-00503-f004:**
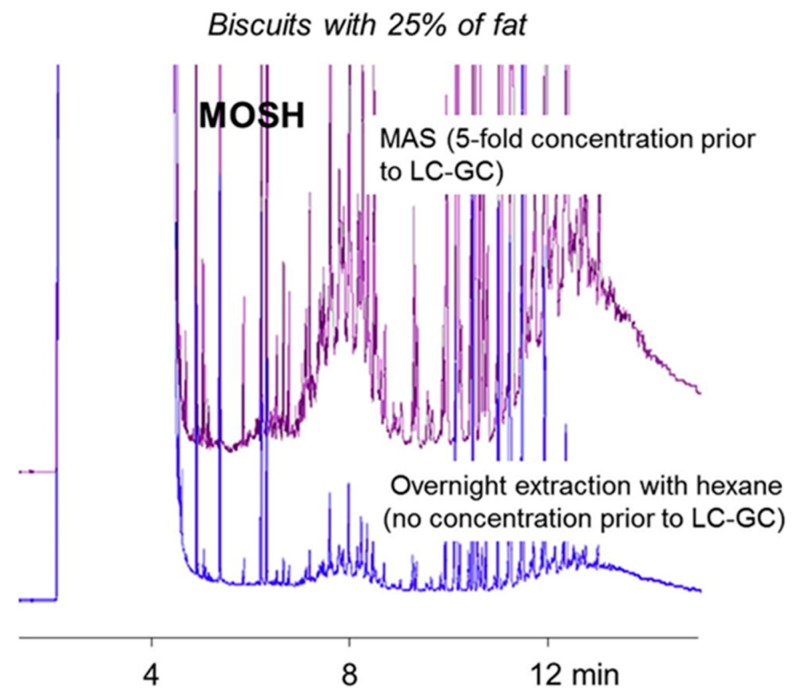
Comparison between MOSH (mineral oil saturated hydrocarbons) traces of a biscuit sample, containing 25% fat, analysed after overnight or after MAS (microwave assisted saponification) and a 5-fold concentration (injecting an amount of extract corresponding to 250 mg of sample containing more than 70 mg of fat Reprinted from Reference [14] with permission from Elsevier.

**Figure 5 foods-08-00503-f005:**
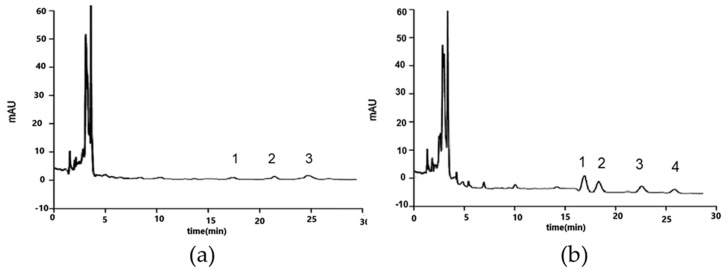
Chromatogram obtained by MAE-DLLME-HPLC-UV (microwave-assisted extraction-dispersive liquid-liquid micro extraction-high performance liquid chromatography-ultraviolet detection) for toasted bread. (**a**) Non-spiked sample (b) Sample spiked with 20 ng/g of PAH4, (1) benzo (a) anthracene; (2), chrysene; (3) benzo (**b**) fluoranthene; (4) benzo (a) pyrene. Reprinted from Reference [51] with permission from Royal Society of Chemistry.

**Table 1 foods-08-00503-t001:** Summary of recent microwave-assisted saponification (MAS) applications regarding hydrocarbon contaminants in food.

Compounds and Food	MAS Condition	Sample Post Treatment	Analysis	Recovery (%)	LODng/g (or mL)	Reference
EPA PAH Pumpkin seed oil	1 g sample + 35 mL of 1.5 M KOH in methanol; microwaved at 130 °C for 10 min	LLE, SPE (on 0.5 g of silica gel)	GC-MS	>83		[39]
Aliphatic hydrocarb PAH Marine biota	20 g sample + 50 mL ethanolic KOH (12%); microwaved at 600 W for 45 min (T max 85 °C)	MAE (3 × 40 mL *n*-hexane at 100 W for 12 min), LLE. SPE on 5 g Florisil	GC-FID	>90, except for lighter PAH		[40]
6 heavy PAH Fish and mussels	1 g of fresh fish (or 0.2 g lyophilized + 0.8 mL of water) + 4 mL of saturated KOH in methanol + 10 mL of *n*-hexane; microwaved at 129 °C for 17 min.	SPE clean-up on silica gel (0.5 g)	HPLC-DAD/FLD	85-100 except for IP, DBahA, BghiP*	0.1–0.5	[41]
10 PAH Fish and mussels	2 g of lyophilized sample + 12 mL saturated methanolic KOH + 12 mL *n*-hexane; microwaved for 15 min at 80% of the power	No sample post-treatment	GC-MS	90–115,		[42]
EPA-PAH Smoked fish and meat	0.4 g of lyophilized sample + 1.6 mL of water + 8 mL of saturated KOH in methanol + 20 mL of *n*-hexane; microwaved at 120 °C for 20 min	SPE clean-up on silica gel (0.5 g)	HPLC-FLD	75–109	<0.1	[13]
EPA-PAH Propolis and propolis- extracts	0.2 g of propolis (or 0.5 g of propolis extract) + 8 mL of a saturated solution of KOH in methanol + 20 mL of *n*-hexane; microw. at 120 °C for 20 min	No sample post-treatment	HPLC-FLD	91–102	<0.1	[43]
EPA-PAH Roasted plants	0.4 g of lyophilized sample + 1.6 mL of water + 8 mL of saturated KOH in methanol + 20 mL of *n*-hexane; microwaved at 120 °C for 20 min	SPE clean-up on silica gel (0.5 g)	HPLC-FLD	<74 (for PAH8)	<0.1	[44]
Mineral oils Cereal based products	5 g sample + I.S.+ 10–20 mL saturated KOH in methanol (depending on the fat content) + 10 mL *n*-hexane; microwaved at 120 °C for 20 min	Rapid wash with water/methanol; follows sample concentration	on-line HPLC-GC	89–104 (MOSH) 85–108 (MOAH)	30	[14]

Abbreviations: LOD: limit of detection; EPA-PAH: Environmental Protection Agency–polycyclic aromatic hydrocarbons; GC-MS: gas chromatography–mass spectrometry; GC-FID: gas-chromatography–flame ionization detection; HPLC: high performance liquid chromatography–photodiode array detector/spectrofluorometric detection; SPE: Solid phase extraction; LLE: liquid-liquid extraction; FLD: spectrofluorometric detection.

**Table 2 foods-08-00503-t002:** Summary of recent studies combining MAE (MAS) with DLLME for determining hydrocarbon contaminants in food.

Compounds and Food	MAE Conditions	Pre-Treatment before DLLME	Extractant/Disperser	Analysis	Enrichment Factors	Recovery (%)	LODng/g (or mL)	Reference
EPA PAH Smoked fish	1 g sample + 12 mL KOH (2 M)/ethanol (50:50)	pH 6.5	Ethylene tetrachloride (100 μL)/Acetone (500 uL)	GC-MS	244–373	82–106	0.11–0.43	[46]
EPA PAH Smoked fish	1 g sample + 10 mL KOH (2 M)/ethanol (50:50); microw. at 500 MHz for 2 min.	pH 5 + Carrez	Ethylene tetrachloride (150 μL)/Acetone (500 uL)	GC-MS				[47]
EPA PAH Grilled meat	1 g sample + 10 mL of KOH (2 M)/ethanol (50:50); microwaved at 500 MHz for 1.5 min	pH 6 + Carrez	Ethylene tetrachloride (80 μL)/Acetone (300 μL)	GC-MS	110–265	85–104	0.15–0.3	[48]
PAH 4Smoked rice	1 g of sample + 10 mL of KOH (2 M)/ethanol 1:1); microwaved at 500 MHz for 1.5 min	pH 5 + Carrez	Chloroform (250 μL)/Acetonitrile (1.2 mL)	HPLC-UV	258–307	87–98	0.05–0.12	[49]
EPA PAH Bread	1 g of sample + 10 mL of KOH (1 M)/ethanol (60:40); microwaved at 500 MHz for 1.5 min	pH 6.5 + Carrez	Ethylene tetrachloride (80 μL)/Acetone (300 μL)	GC-MS	200–300	85–104	0.1–0.3	[50]
PAH4 Toasted bread	2 g of sample + 10 mL of KOH (1 M)/ethanol (50:50); microwaved at 500 MHz for 1.5 min	pH 6 + Carrez	Chloroform (180 μL)/Acetonitrile (0.9 mL)	HPLC	255–312	87–104	0.03–0.19	[51]
Heavy PAH Coffee	0.5 g sample + 0.5 mLwater + 10 mL KOH (85% *v*/*v*) in ethanol; microw. at 520 MHz for 8 min	pH 6 + Carrez	Ethylene tetrachloride (80 μL)/Acetone (300 μL)	GC-MS	155–248	88–101	0.1–0.3	[52]
PAH vegetables	10 mL homogenate sample + 4 mL acetone; microw. at 400 W for 1.5 min		1-bromo-3-methylbutane (30 µL)/Acetone (800 µL)	GC-FID				[53]
PAH Edible oils	Sample + acetonitrile/acetone (1:1) and methanolic KOH (in two steps).		Ethylene tetrachloride/Ethanol	GC-MS	81–124		0.2–2.7	[54]

Abbreviations: DLLME: dispersive liquid-liquid micro-extraction; UV: ultraviolet.

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
