# Peer review of "Microwave-Based Technique for Fast and Reliable Extraction of Organic Contaminants from Food, with a Special Focus on Hydrocarbon Contaminants"

_foods, 2019, doi:10.3390/foods8100503_

Round 1
Reviewer 1 Report
The authors present a review where expose the analytical procedures based on microwave assisted extraction (MAE) for the determination of polycyclic aromatic and mineral oil hydrocarbon compounds in food.
The article is well written and the reference list is enough for a review. Moreover, the details of the references used are appropriate.
Nevertheless, I suggest some minor changes in the manuscript:
Add the laws and regulations about polycyclic aromatic and mineral oil hydrocarbon compounds to the reference list. Lines 89 - 92. Please, revise the sentence. It is not totally clear which type of microwave-assisted extraction correspond to open and close vessels. Section 2: Please, consider to add a small paragraph about the differences and advantages/disadvantages of the heating using microwaves in comparison with traditional heating. Section 3: ¿Which of these systems are more common nowadays? Lines 168-176: Please provide some references for the explanation of the extraction modes. Line 207: Indicate the different techniques used as purification steps. Line 271: It is not necessary to indicate again which means “DLLME”. Line 315: Microwave-assisted micellar extraction (MAME) Table 1, Table 2: Turn the tittles because it is quite difficult to read in the present form.
Author Response
The authors thanks the referee for the work done.
Concerning the suggestion to add a paragraph about the differences and advantage/disadvantages of the heating using microwaves in comparison with traditional heating, paragraph 118-122 (which already reported these information) has been partially modified.
Concerning the suggestion to turn the title of table 1 and 2, it has been partially done, where possible. Due to space problems some title remained in the vertical format.
All other suggestions have been accepted and the text modified accordingly.
Reviewer 2 Report
This review article discusses the use of microwave-based techniques for the extraction of organic contaminants from food, particularly, polycyclic aromatic hydrocarbons (PAHs) and mineral oil hydrocarbons (MOH). The manuscript is generally well written and organized, focusing the main aspects of this technique (principle, equipment, extraction mode, sample processing previous to extraction, parameters that affect sample extraction, sample clean-up, etc.).
The more recent use of Microwave-Assisted Saponification (MAS) and the utilization of ionic liquids in MAE is also presented. The references were selected from scientific papers published in the last 10 years, what makes the review up-to date.
The topic chosen is very important and useful, and the information presented is critically discussed.
Therefore, I recommend the publication of this manuscript after minor revision.
Specific comments:
Line 34 “The First application…” “The first application…”
Line 61 “…contrarily from…” “…contrarily to…”
Line 80 “…traditional saponification etc.)…” “…traditional saponification, etc.)…”
Line81 “…it is time and solvent consuming,…” “…it is time consuming,…”
Line 85 “…in sample preparations,…” “…in sample preparation,…”
Lines 109 and 110 Mhz MHz
Line 138 “Extraction cell…” “Extraction cells…”
Line 150 “…enable…” “…enables…”
Line 214 “…enlarging the range of co-extracted.” “…enlarging the range of co-extracted compounds.”
Lines 224 and 225 I suggest removing “(“ and “)”
Line 239 “..has been…” “..have been…”
Line 318 “(reverse phase)” “(reversed phase)”
Line 341 “…lower of…” “…lower than…”
Line 371 “Table 1, summarize…” “Table 1 summarizes…”
Table 1 Why is reference [42] cited firstly than [41]?
Lines 419 and 420 and also in the Abbreviations Two different designations for the same compound F – fluorene ; Fl - fluorene
One of the abbreviations can probably refer to Fluoranthene. Please check.
Line 453 “DLLE” “DLLME”
Line 443 “…of fa…” “…of fat…”
Table 2, column 5 header “Analytsis” “Analysis”
Line 470 “…select ion monitoring (SIM)…” “…selected ion monitoring (SIM)…”
Line 475 “(table 1)” “(table 2)”
Line 483 “(50:50 v/v)” “(50:50, v/v)”
Line 524 “…microwave irradiation…” “…microwave radiation…”
Line 555 “…prevails by far on FMAE.” “…prevails by far over FMAE.”
Line 576 “…abbreviation are…” “…abbreviations are…”
RP reversed phase
SIM selected ion monitoring
Author Response
The authors thanks the reviewer for the careful check of the paper.
All the modification/corrections suggested by the reviewer have been accepted.
Citations of table 1 (some of which already reported in the text), have been ordered according to the publication year (from the less recent to the most recent).